# Self-Reported Questionnaire to Evaluate Functional Abilities in Middle Age: A Call for Delphi Expert Panel

**DOI:** 10.3390/healthcare11071040

**Published:** 2023-04-04

**Authors:** Roee Hayek, Odelyah Saad, Shmuel Springer

**Affiliations:** 1The Neuromuscular & Human Performance Laboratory, Department of Physical Therapy, Faculty of Health Sciences, Ariel University, Ariel 40700, Israel; 2Department of Nursing, Faculty of Health Science, Ariel University, Ariel 40700, Israel

**Keywords:** middle age, aging, self-assessment, self-reported, function, questionnaire

## Abstract

Early detection of functional decline is important for promoting optimal aging. Self-reported questionnaires can efficiently assess functional abilities. Therefore, we aimed to highlight the lack of functional ability assessment questionnaires for the middle-aged population (MA, 45–65 years) and the need to develop such a questionnaire. An online search was performed to find questionnaires quantifying self-reported performance and functional abilities at MA. We also conducted an online survey of a group of the MA population and interviewed individuals who reported age-related functional decline. Eight potentially relevant questionnaires were found, and one hundred and twenty-three individuals responded to our survey, five of whom were interviewed. None of the questionnaires were specifically designed to assess functional capacity at MA, and most of the questionnaires are likely to have a ceiling effect in assessing the MA population. Furthermore, the questionnaires do not capture functions related to dynamic balance, flexibility, and maximum strength, which are reported as difficult by our respondents, making them less appropriate for assessing function at MA. There is a need to engage a Delphi expert panel of several relevant healthcare professionals to develop a functional capacity assessment questionnaire for MA.

## 1. Introduction

Middle age (MA, 45–65 years) is an important stage in the life course, linking young and late adulthood to old age. At this age, changes in body systems such as the neuromuscular, somatosensory, and vestibular systems occur and declines in various functions can be observed [1,2]. Evidence suggests that impairments in balance and locomotion are widespread in MA [3,4], and that falls related to slips and trips are common in this population [5]. Early identification of functional impairments in MA may be important to implement appropriate interventions and initiatives to increase physical activity and exercise, which have been shown to have a positive impact on the aging process [6]. Due to the wide age range, differences in lifestyle habits, and development of metabolic syndromes, variance in functional abilities is to be expected at MA, even in people who have only mild and general health problems [4]. However, most current assessment methods may not adequately capture differences in functional abilities within the MA population [7,8].

Self-reported questionnaires are commonly used for functional assessment because they provide a cost-effective and efficient way to collect data [9,10]. Physical functional status refers to a person’s ability to perform basic, instrumental, and advanced activities of daily living. Advantages of using questionnaires include the ability to administer them to a large number of participants in a relatively short time, the ability to distribute them remotely, standardized data collection, and relatively low cost [9,10,11]. However, there are some disadvantages to using questionnaires for functional assessment. Participants may not fully understand the questions or answer them correctly, and they may also be unaware of their personal capabilities, resulting in incomplete or inaccurate data [10,11]. Self-reported questionnaires can be an effective tool for functional assessment at MA, but should be designed to address their known weaknesses. To the best of our knowledge, no questionnaire assesses self-reported functional abilities in MA.

Therefore, this work aimed to demonstrate the lack of functional ability assessment questionnaires for the middle-aged population and the need for a functional ability assessment questionnaire specifically designed for MA.

## 2. Materials and Methods

A search for an appropriate questionnaire for functional assessment in this age group was conducted by three researchers. The search was based on the English keywords used individually or in combination (according to the MeSH): “questionnaire”, “survey”, “middle age”, “Activity of Daily Living (ADL)”, “assessment”, “evaluation”, “aging”, “function”, “basic functions”, “self-care”, “mobility”, “physical activity”, “rehabilitation”. Searches were conducted in English database sources, including Medlin, PubMed, Google Scholar, Cochrane, ProQuest, Web of Science, and Pedro. Eligibility criteria for relevant questionnaires included questionnaires that quantify self-reported performance and functional ability without a high probability of a ceiling effect in MA. For example, one of the most commonly used functional measures in the elderly in both research and clinical settings is the Barthel Index [12], which includes ten functional domains (bowels, bladder, personal hygiene, toileting, feeding, transfer, mobility, dressing, stair climbing, and bathing). Response options are summed for a total score ranging from zero (low function, dependent) to 100 (high function, independent). As the functional domains only provide information about basic activities of daily living, it is likely that most in the MA population would score very high.

To better understand the extent of functional decline at MA, and how and when it manifests, we also conducted an anonymous online open survey among a group of MA adults. The survey was posted on social media platforms and WhatsApp groups to family, friends, work colleagues, and anyone in the appropriate age group. Our goal was to obtain one hundred responses to gain more insight into the prevalence of the phenomenon. The survey included the following questions: (1) Do you feel a change in performing functions in your daily life compared to your level of functioning a few years ago? If so, in what functions? (2) Do you feel that your ability to participate in sports and recreational activities (including cardiopulmonary endurance) has changed compared to your ability several years ago? (3) Do you feel that your physical abilities in terms of strength, flexibility, stability, and balance have changed compared to younger adulthood?

We also interviewed five MA individuals who reported age-related functional decline. Participants in the online survey were asked to contact the researchers if they were concerned about a negative change in function. For this purpose, the questionnaire included the researchers’ contact information. The interview was conducted by telephone without asking the respondent for his or her personal information. Our goal was to interview five of the planned 100 survey participants. Interview topics addressed multiple levels of functioning, such as (1) instrumental ADLs (IADLs) including organizational skills such as housekeeping, shopping, transportation, etc.; (2) advanced ADLs (AADLs) such as hobbies and work, travel and outings, or caring for a grandchild; (3) moderate to vigorous physical activity; another category we referred to was (4) any difficulty with general movement, such as getting up from sitting, bending, flexibility, and stiffness.

## 3. Results

The search yielded eight potentially relevant questionnaires, including (1) the physical functioning (10 items) of the short-form survey (SF-36) [13], a self-reported measure of physical health. This item measures the individual’s ability to perform physical activities such as climbing stairs, carrying groceries, and walking short distances. (2) The Norwegian Function Assessment Scale (NFAS) [14,15], a self-report instrument based on the International Classification of Functioning, Disability, and Health (ICF). It consists of 10 items that cover various aspects of physical function, including mobility, self-care, and work-related activities. (3) The Patient-Reported Outcomes Measurement Information System (PROMIS^®^) [16], a set of person-centered measures that evaluates and monitors physical, mental, and social health in adults and children. The PROMIS^®^ includes a large number of item banks, or sets of items, that assess various aspects of health, including physical function. The item banks are designed to be highly responsive to changes in an individual’s health status, and they use computer algorithms to select the most appropriate items for each individual based on their responses. (4) The self-reported physical fitness survey (SRFit) [17,18], an instrument that measures an individual’s (adults ≥ 40 years of age) perceived level of physical fitness. It consists of items that assess various aspects of physical fitness, including cardiovascular endurance, muscular strength, flexibility, and body composition, among others. Although they do not exactly meet the eligibility criteria, the following questionnaires were also considered and evaluated, because they have been previously used to study MA populations: (5) The International Fitness Scale (IFIS) [19]. This questionnaire was developed by Ortego et al. (2011) [19] to study adolescents and provides a measure of fitness based on responses to five basic fitness questions, with responses based on a five-point Likert scale. (6) The perceived physical fitness scale (PPFS) [20], an instrument designed to assess individual perceptions of physical fitness. (7) Quick physical activity rating (QPAR) [21], an informant-rated instrument to quantify physical activity levels. (8) CLINIMEX - Aerobic Fitness Questionnaire [22], a tool to estimate aerobic fitness with corresponding values in metabolic equivalents (METs).

One hundred twenty-three MA people responded to the survey within a week. Of the respondents, only 9% (11 people) indicated that their functional abilities had improved with age, mainly thanks to the fact that they have been involved in physical activities in recent years, 31% (36 people) indicated that their functional abilities had not changed, while most respondents, 60% (76 people), noted some deterioration in their functions. The functions most frequently reported to have deteriorated were travel and hiking (10%, 13 persons), household (12%, 15 persons), and stair climbing (17%, 21 persons). In addition, 8% (10 individuals) reported functional limitations due to pain and 11% (14 individuals) reported increased fatigue after normal daily activities. In terms of vigorous activity, 33% reported much more difficulty recovering after an exercise or an effort than before. Decreased flexibility was reported by 23% of the subjects and 12% reported reduction in balance. Three out of the five respondents who were interviewed due to age-related functional decline reported difficulty with IDAL “I have a lot of difficulty getting up and down the stairs, doing housework like cooking and cleaning, and doing anything that requires physical effort”. Two of five reported difficulty with AADL: “About five years ago, in my late 50s, I had difficulty with challenging walking distances (trips); now, at 56, I do not do trips that require challenging walking”. “Walking after grandchildren has become a little more difficult”. “I do not walk with my wife anymore because I have a feeling of heaviness and fatigue”. They also reported that they no longer engage in intense physical activities due to physical difficulties. Two respondents reported having difficulty only with moderate to vigorous physical activity: “The intensity of the workout at the gym is lower”. “I cannot participate in sports classes that are more active and intense”. “Running times for the same distance have become longer, it is harder to recover from runs than before”. All respondents reported difficulty with general movements: “Bending over the closet and standing up is much harder than it used to be, I feel like my knees do not have strength anymore”. “Bending down with my knees”. “I feel less mobile and cannot run because my body is stiff”. “It’s hard for me to bend over to pick something up off the floor and get back up”. “Poor balance, I slipped once and I think that is the reason”. In addition, four out of five respondents said they suffer from lower back, knee, and hip pain.

## 4. Discussion

### 4.1. Potentially Relevant Questionnaires

Two of the eight potentially relevant questionnaires, the physical functioning portion of the SF-36 [13,23,24,25], and the NFAS [14,26,27], are commonly used to assess the functional abilities of MA adults. The Midlife in the United States (MIDUS), a national longitudinal health and well-being study (http://midus.wisc.edu/ accessed on 1 February 2023), evaluated functional health by using the physical functioning subscale of the SF-36 [28]. The NFAS was used in the 2011/12 workplace mental health study to assess the functional ability of 3937 workers in Germany aged 31–60 years [26]. However, both questionnaires may not be appropriate for assessing function in MA adults. Some questions on the SF-36 are likely to have a ceiling effect (e.g., questions 9–12: “Does your health now limit you in walking one block?” or “Does your health now limit you in bathing or dressing?”). In addition, the questionnaire does not capture functions requiring balance, flexibility, and maximum strength, which may be impaired in MA. The NFAS may be more comprehensive than the SF-36 for functional assessment, but it also does not address balance and flexibility, intense aerobic activity, and outdoor recreation. Moreover, even in the NFAS, some of the questions have a clear ceiling effect for MA people, such as those asking about difficulty standing, putting on shoes and socks, sitting on a kitchen chair, or dressing and undressing. Finally, the physical portion of the SF-36 includes a three-point Likert scale that is considered less accurate for assessing functional abilities, as it is argued that the minimum number of categories should be in the range of five to seven [29,30,31]. The NFAS was originally designed with a four-point scale, but a five-point scale was later developed. A comparison between the four-point and five-point scales of the NFAS revealed that the five-point Likert scale had better data quality and higher internal consistency [27]. Thus, although both questionnaires have been used to examine function in MA people, they appear to have several disadvantages that make them less suitable for assessing function in this age group. Therefore, a more targeted questionnaire is needed for assessing function in this age group.

Appropriate questions for assessing physical function at MA can be obtained from PROMIS^®^ [16,32]. The PROMIS is a large pool of questions from which many short forms have been developed. These forms serve different purposes, with some of them focusing on self-assessment of ADL skills. Currently, there is no short form comprehensive enough for the assessment of MA adults. However, it is possible that the database of PROMIS questions could be used to create a questionnaire that tests and differentiates the functional abilities of various MA individuals.

The SRFit [17], designed for MA adults and containing items on strength, endurance, and flexibility, may not be appropriate for our purpose because it focuses on specific functions that the subject must perform in many cases to test his or her abilities (because they are very specific). Moreover, the questionnaire contains only a few tasks from daily life, without advanced functions. Other potentially relevant questionnaires such as the IFIS [19] and the PPFS [20] assess self-perceived general ability compared to others of the same age and do not quantify the subject’s ability. In addition, they have not been validated for MA adults.

Physical activity questionnaires such as the QPAR [21] have also been adapted for older adults. However, these types of questionnaires examine the frequency of performing certain actions rather than the person’s ability to perform them.

The last potentially relevant questionnaire is the CLINIMEX [22], which focuses on aerobic fitness. Although the questionnaire was tested on MA adults, it does not fully meet our objective because it focuses specifically on aerobic capacity and does not test self-reported function.

### 4.2. The Online Survey and Interview

The survey responses show how variable functional abilities can be at this stage of life. While some respondents reported no deterioration and even improvement in their functional abilities, most respondents (60%) reported deterioration in their functional abilities. This manifested itself in a variety of functions and activities, from changes in functions necessary for daily living to changes in intense athletic activities. By interviewing five individuals, we were able to deepen our understanding of the difficulties and functional declines experienced by MA adults. We recognized that travel and walking, functions required for household chores, and stair climbing are among the functions that require the most attention in this population. We also found that flexibility, strength, and balance tend to decrease at MA and may lead to difficulties in daily living. In addition, some MA individuals may experience impairment only during high-intensity activities, such as sports or physical training, and this also requires attention.

This work should be interpreted with some limitations. We used an open survey rather than a validated questionnaire because the existing questionnaires are not specific for evaluating function at MA, as described in this work. Also, because our goal was to gain general insight into the prevalence of functional decline in MA, regardless of health status, we did not ask about the health status of participants, although it is likely that the population of MA is affected by several medical conditions that may affect function [4]. Therefore, future studies assessing physical function in the MA population with an appropriately validated questionnaire and taking into account variables such as health status are warranted.

## 5. Conclusions

The need for a functional assessment questionnaire specifically designed for MA is critical, as this population faces unique challenges and health issues. While several functional assessment questionnaires exist, none have been developed specifically for assessing function at MA. In addition, the existing questionnaires have several drawbacks that make them less appropriate for assessing function at MA. A Delphi expert panel composed of several relevant health professionals should be convened to develop a functional assessment questionnaire tailored to the specific needs and concerns of this age group. The newly developed questionnaire can combine questions from different physical assessment instruments. It should be designed to capture a broad range of abilities and health measures relevant to MA. These include (1) IADL, (2) AADL, (3) general strength, (4) basic to intensive aerobic capacity, (5) flexibility, (6) balance, and (7) general health. Such a questionnaire will not only provide valuable information for healthcare providers, but will also aid in the early detection and treatment of age-related health issues in this population.

## Data Availability

The data are available from the corresponding author on reasonable request.

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
