# Peer review of "Self-Reported Questionnaire to Evaluate Functional Abilities in Middle Age: A Call for Delphi Expert Panel"

_healthcare, 2023, doi:10.3390/healthcare11071040_

Round 1
Reviewer 1 Report
Thank you for inviting me to review this manuscript on “Self-reported questionnaire to evaluate Functional abilities in middle age: a call for Delphi expert panel.“
I think the topic is very important for readers in Healthcare. There are some concerns in the study.
1. Using the 8 questionnaires presented, have you interviewed 123 people? If so, showing the distribution of the actual scores and the ceiling effect would strengthen your argument.
2. Please describe in detail how the 123 people who answered the questions and the 5 people who responded to the interviews were selected.
Author Response
Reviewer #1:
I think the topic is very important for readers in Healthcare. There are some concerns in the study.
Response: Thank you for the review and comments.
- Using the 8 questionnaires presented, have you interviewed 123 people? If so, showing the distribution of the actual scores and the ceiling effect would strengthen your argument.
Response: The 123 people who responded to our survey were not asked to complete the eight available questionnaires. For the survey, we used an open-ended questionnaire to gain more insight into the prevalence of functional decline in middle age. In accordance with the reviewer's comment and to further clarify this issue, we refined the methods section and added a sentence to the limitations section stating that we did not use valid questionnaires to monitor functional decline in middle-aged people.
- Please describe in detail how the 123 people who answered the questions and the 5 people who responded to the interviews were selected.
Response: This was done as suggested. Based on this comment, we have expanded the description of the selection of participants in the Methods section. Our goal was to recruit one hundred participants (i.e., responders). Five participants contacted the authors to conduct the telephone interview, and over time 123 people responded to the survey; thus, this was the sample used for data analysis.
Reviewer 2 Report
Dear Scientists and Editor; Considering the increasing loss of functional abilities with aging, your study is quite significant.Congratulations on your preparation for the survey and conceptual framework of the study.
I recommended you mention the positive effects of physical activities and exercise on the aging process in the introduction part.
Additionally please indicate and clarify how many people you reached to participate in the survey and how many accept it at the methods part.
Finally, do you have any information about the participants' medical history?
Author Response
Reviewer #2:
- I recommended you mention the positive effects of physical activities and exercise on the aging process in the introduction part.
Response: Thank you for this important note, it has been added as suggested.
- Indicate and clarify how many people you reached to participate in the survey and how many accept it at the methods.
Response: Thank you for your comment. This point has been clarified in the revised methods section. Our goal was to collect one hundred responses to the survey we conducted and five telephone interviews. To that end, we contacted groups on social media and via WhatsApp. We do not know how many people the survey reached, but we found that we were able to conduct five telephone interviews within a week and that 123 people responded to the survey during that period.
- Do you have any information about the participants' medical history?
Response: We did not ask respondents about their health status because our goal was to gain general insight into the prevalence of functional decline in middle age, regardless of health status. Nevertheless, based on this comment, we have added a limitation to the discussion section indicating that health status may influence respondents' functional level.
Round 2
Reviewer 1 Report
The author has responded carefully to the questions I raised point by point.